# Structural basis for high-affinity actin binding revealed by a β-III-spectrin SCA5 missense mutation

Adam W. Avery[1], Michael E. Fealey [2], Fengbin Wang [3], Albina Orlova[3], Andrew R. Thompson [2], David D. Thomas [2], Thomas S. Hays[1] & Edward H. Egelman [3]

Spinocerebellar ataxia type 5 (SCA5) is a neurodegenerative disease caused by mutations in the cytoskeletal protein β-III-spectrin. Previously, a SCA5 mutation resulting in a leucine-to-proline substitution (L253P) in the actin-binding domain (ABD) was shown to cause a 1000-fold increase in actin-binding affinity. However, the structural basis for this increase is unknown. Here, we report a 6.9 Å cryo-EM structure of F-actin complexed with the L253P ABD. This structure, along with co-sedimentation and pulsed-EPR measurements, demonstrates that high-affinity binding caused by the CH2-localized mutation is due to opening of the two CH domains. This enables CH1 to bind actin aided by an unstructured N-terminal region that becomes α-helical upon binding. This helix is required for association with actin as truncation eliminates binding. Collectively, these results shed light on the mechanism by which β-III-spectrin, and likely similar actin-binding proteins, interact with actin, and how this mechanism can be perturbed to cause disease.

[1] Department of Genetics, Cell Biology and Development, University of Minnesota, Minneapolis, MN 55455, USA. [2] Department of Biochemistry, Molecular Biology and Biophysics, University of Minnesota, Minneapolis, MN 55455, USA. [3] Department of Biochemistry and Molecular Genetics, University of Virginia, Charlottesville, VA 22908, USA. Thomas S. Hays and Edward H. Egelman jointly supervised this work. Correspondence and requests for materials should be addressed to T.S.H. (email: haysx001@umn.edu)

Spinocerebellar ataxia type 5 (SCA5) is a neurodegenerative disease that stems from autosomal dominant mutations in the cytoskeletal protein β-III-spectrin[1, 2]. SCA5 pathogenesis results from a functional deficit in Purkinje cells, in which the expression of β-III-spectrin is required for normal cerebellar control of motor coordination[3]. β-III-spectrin is thought to form a heterotetrameric complex with α-II-spectrin, and to cross-link actin filaments to form a cytoskeleton localizing to the shafts and spines of Purkinje cell dendrites. β-III-spectrin is required for normal dendrite structure[4] and synaptic transmission[5, 6]. Recently, our group reported that a SCA5 missense mutation, L253P, localized to the β-III-spectrin N-terminal actin-binding domain (ABD), causes a ~1000-fold increase in actin-binding affinity[7]. Here, we probe the structural mechanism of this mutation by studying the ABD with complementary biophysical techniques.

The β-III-spectrin ABD comprises tandem calponin homology (CH) domains and the L253P mutation is localized to the second subdomain (CH2). Very little is known about the structural biology of β-III-spectrin's ABD, with the closest related atomic model being the isolated CH2 domain of β-II-spectrin[8]. Crystal structures of N-terminal ABDs from the spectrin superfamily, including α-actinin, dystrophin and utrophin, invariably show that extensive contacts are made between CH1 and CH2, suggesting a tendency to exist in a "closed" conformation in the absence of actin[9–12]. A cryo-EM structure of the fimbrin ABD shows that it associates with actin in a closed structural state[13]. In contrast, cryo-EM showed that α-actinin associates with actin in an "open" structural state in which only a single CH domain is bound to the filament and the second domain is structurally

disordered on account of it being dissociated from the interacting CH domain[14]. A similar conclusion was reached for filamin[15], another member of the spectrin superfamily. Binding studies suggested that the CH1 domain of α-actinin has greater intrinsic affinity for actin in isolation[16] and this suggested that it was CH1 bound in the cryo-EM structure. This led to the hypothesis that the CH2 domain functions to regulate the actin-binding function of CH1 through steric hindrance when the two domains are associated. Consistent with this, many mutations in the CH2 domains of both α-actinin and filamin impart modest gains in ABD affinity for actin[17, 18]. Collectively, these studies suggest that the L253P mutation of β-III-spectrin, which is similarly localized to CH2, causes high-affinity actin binding by disrupting a regulatory mechanism that shifts the ABD structural equilibrium from a closed to more open binding-competent state. Here, we report cryo-EM, co-sedimentation, and pulsed electron paramagnetic resonance (EPR) data consistent with such a mechanism.

## Results

**Structure of L253P β-III-spectrin ABD bound to actin.** To begin testing our hypothesis, we first performed cryo-EM on the β-III-spectrin ABD bound to actin filaments. The actin-binding affinity of the wild-type (WT) β-III-spectrin ABD is low ($K_d = 75$ μM), resulting in poorly decorated actin filaments that were of insufficient quality for analysis (Supplementary Fig. 1a). The L253P ABD yielded high-quality complexes of decorated filaments (Supplementary Fig. 1b, c), enabling a three-dimensional reconstruction (Fig. 1a) of the mutant ABD–actin complex at 6.9

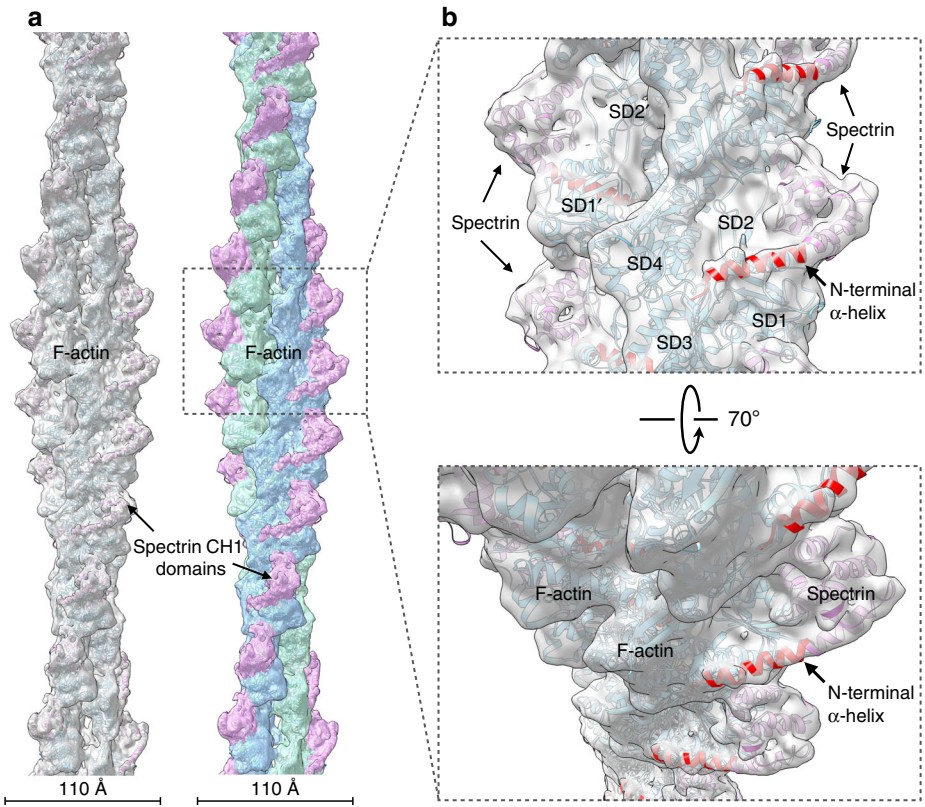

**Fig. 1** Cryo-EM map and model of L253P β-III-spectrin ABD bound to actin. **a** The map (left, gray transparent surface) has been fit with a model for actin (cyan) and the β-III-spectrin ABD (magenta). On the right, the surface of the reconstruction has been color coded for the two actin strands (blue and green) and the β-III-spectrin ABD (magenta). **b** Close-up view of **a** showing that the CH1 domain has an additional N-terminal helix (red) interacting with F-actin. The actin subdomains (SD1, SD2, SD3, and SD4) have been labeled on one actin subunit, while SD1′ and SD2′ are labeled on a different subunit

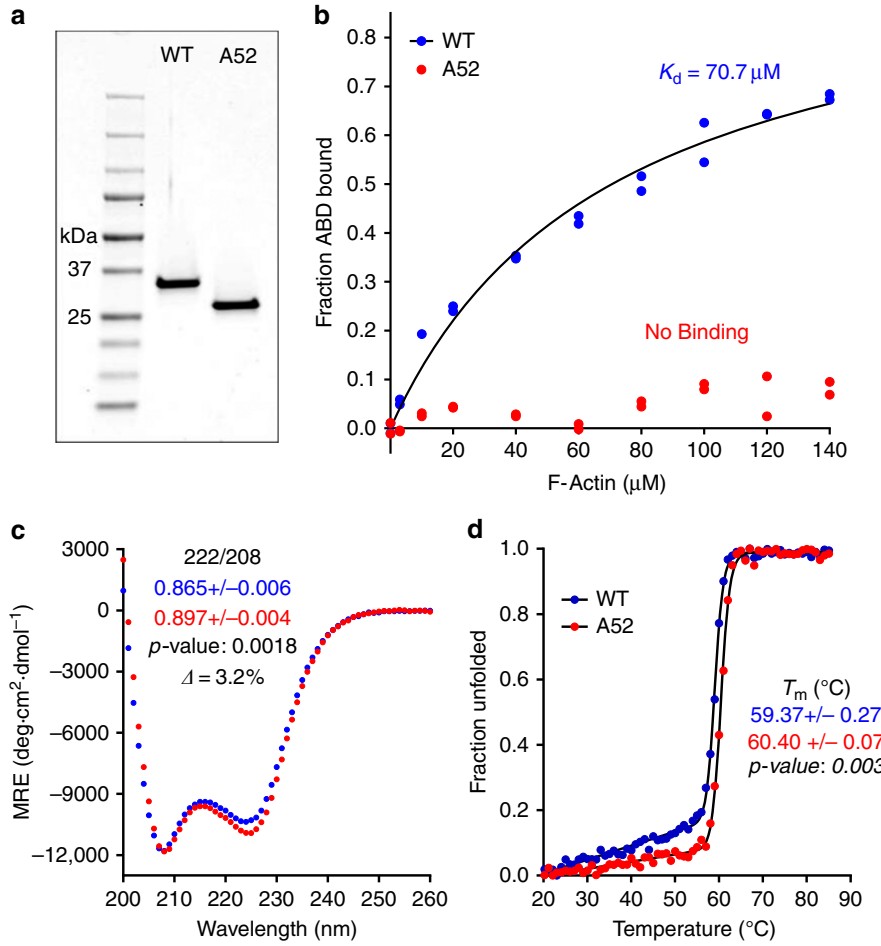

**Fig. 2** The β-III-spectrin N-terminus is required for actin binding. **a** Coomassie blue stained gel of purified WT ABD or WT ABD without the N-terminal 51 amino acids (A52). **b** F-actin co-sedimentation assays showing that the N-terminal truncation abolishes actin affinity. **c** CD spectra demonstrating α-helical absorption profiles. The A52 ABD has a statistically significant increase in helicity (n = 3). **d** CD denaturation at 222 nm. The A52 ABD has a statistically significant increase in $T_m$ (n = 3)

Å resolution (Supplementary Fig. 2). This represents a substantial improvement over previous ABD–actin reconstructions[13, 14, 19], the best of which was 12 Å. The reconstruction and resulting atomic model provide several mechanistic insights. First, the density map reveals that only a single CH domain is bound to actin, as observed previously for the α-actinin ABD. Second, the bound CH domain has an additional N-terminal helix that is tightly associated with actin (Fig. 1b, red). This helix was not identified in other ABD–actin cryo-EM complexes. However, reexamination of the α-actinin–actin reconstruction[14] suggests that extra density is present, consistent with such an N-terminal helical extension. By comparison, the higher resolution fimbrin–actin reconstruction[13], containing closed CH domains, shows no extra density. The presence of this contiguous N-terminal helix unambiguously identifies the bound domain as CH1. Thus, high-affinity actin binding, caused by the L253P mutation in the CH2 domain, is mediated through the CH1 domain. The L253P mutation does not expose or generate a de novo high-affinity actin binding site in the CH2 domain, as has been suggested previously[20].

All N-terminal ABDs contain amino acid sequences of variable length and composition preceding the conserved CH1 domain. However, a structured N-terminal region preceding the globular fold of a CH domain has not been previously observed in most reported ABD crystal structures. This reflects either disorder in this region or the intentional truncation of the region based on

predicted intrinsic disorder[21]. However, when calmodulin was crystallized with the plectin ABD, calmodulin was bound to the N-terminal region which had become α-helical[21]. Solution studies confirmed that in the absence of calmodulin, the plectin N-terminal region is unstructured. The β-III-spectrin CH1 domain with the extended N-terminal helix built into the cryo-EM map superimposes very well with the corresponding plectin CH1 domain with calmodulin (Supplementary Fig. 3), and shows that calmodulin would be involved in massive clashes with actin. As proposed this explains how calmodulin, in the presence of $Ca^{2+}$, dissociates the plectin ABD from actin since the binding of actin and calmodulin is competitive[22].

**N-terminal ABD residues are essential for actin binding**. The β-III-spectrin cryo-EM structure showing the N-terminal helix bound to actin suggests that the helix must contribute to binding affinity. To test this, we measured affinity of WT ABD with and without the first 51 amino acids (A52). Strikingly, truncation of the N-terminal sequence abolished binding of the ABD to actin (Fig. 2a, b). Circular dichroism (CD) indicates that this loss in binding is not due to misfolding (Fig. 2c, d). On the contrary, the A52 ABD showed small but reproducible increases in helicity and stability, suggesting that the N-terminal residues contain intrinsic disorder, which we verified by CD (Supplementary Fig. 4). Collectively, these data sets, combined with the cryo-EM

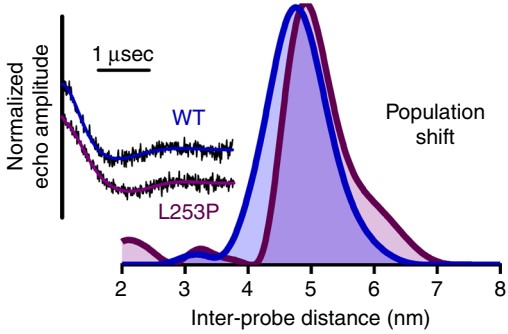

**Fig. 3** DEER measurement showing the L253P mutation opens the β-III-spectrin ABD structure. Echo amplitude decays of WT ABD (blue) and L253P ABD (purple) along with their corresponding Tikhonov fits are shown on the left. The inter-probe distances derived from Tikhonov regularization (Supplementary Fig. 6) for both WT and L253P ABDs are shown on the right. The WT ABD distance distribution is centered at 4.8 nm, consistent with the distance predicted in the homology model of the closed state shown in Supplementary Fig. 5. Upon introduction of the L253P mutation, the distance distribution undergoes a shift to populate a longer inter-probe distance, visible as a shoulder to the right of the 4.8 nm peak, consistent with structural opening of the ABD

structural model, indicate that the N-terminal sequence is critical to binding, both directly by interacting with actin and potentially indirectly through allosteric destabilization of the ABD, the latter of which may alter the ABD conformational ensemble to impact binding. Evidence supporting an allosteric contribution comes from recent folding and binding studies on the utrophin ABD1, which showed that the utrophin N-terminal residues destabilize ABD1 and are required for full ABD1 binding activity. The utrophin N-terminal residues alone do not bind actin[23].

**L253P mutation promotes structural opening of CH domains.** Previously we demonstrated that the L253P mutation substantially destabilizes the ABD ($\Delta T_m = -14.8\,°C$). If the open structural state of the ABD is responsible for high-affinity actin binding, then decreased stability may facilitate opening of the CH domains from a closed state. In our previous structural homology model of the β-III-spectrin ABD in the closed structural state[7], L253 is positioned at the CH domain interface (Supplementary Fig. 5), suggesting that the L253P mutation could also perturb CH1–CH2 interactions that stabilize the closed structural state. To test this hypothesis, we used double electron–electron resonance (DEER) to measure inter-CH domain distance with and without the L253P mutation. We exploited the native cysteine residues at positions 76 and 231 for irreversible attachment of spin labels.

For WT ABD in the absence of actin, clear oscillations were present in the echo amplitude decay (Fig. 3). Analysis (Supplementary Fig. 6) revealed an inter-probe distance centered at 4.8 nm (Fig. 3), which agrees well with the distance predicted for the closed structural state (Supplementary Fig. 5). Upon introduction of the L253P mutation, the distance distribution undergoes a shift to populate a longer inter-probe distance, visible as a shoulder to the right of the 4.8 nm peak. This indicates that the ABD undergoes an opening as a result of the mutation (Fig. 3), consistent with the cryo-EM structure showing that CH1 separates from CH2 upon associating with actin (Fig. 1).

In the context of other ABDs harboring disease-causing mutations, a similar structural mechanism has been proposed for the K255E mutation of α-actinin-4, a missense mutation also located at the CH domain interface. This mutation caused increased affinity for actin, but the crystal structure showed the

ABD of mutant α-actinin-4 to be in a closed state[24]. While seemingly counter to our proposed mechanism here, the DEER-derived distance distributions indicate that the shift to an open conformer is not complete (Fig. 3). Only a small portion of the population exists in the more open state in the absence of actin, with the remaining ensemble occupying the closed conformer. Given that crystallographic conditions favor more stable conformers, the K255E mutant may open, similar to L253P β-III-spectrin, but crystallize in its more stable closed state because it is more significantly populated.

## Discussion

SCA5 pathology is characterized by atrophy of the cerebellum[25], likely reflecting degeneration of dendritic arbors extended by Purkinje cells. Within dendrites, β-III-spectrin binds to actin filaments to form a spectrin-actin skeleton underlying the plasma membrane[26]. The low affinity of WT β-III-spectrin for actin suggests that normal membrane function requires a dynamic spectrin-actin cytoskeleton in which spectrin-actin linkages form and dissociate. We suggest that the high affinity of L253P β-III-spectrin for actin decreases dynamics of spectrin-actin linkages, resulting in reduced plasticity of the spectrin-actin cytoskeleton. We speculate that spectrin-actin cytoskeleton plasticity is important for the cytoskeleton to expand or retract within structurally dynamic regions of the dendritic arbor, such as growing or remodeling dendrites and spines. Recent work has highlighted the requirement of β-III-spectrin to support dynamic spine structure and post-synaptic signaling[6]. In addition, disrupted microtubule-based transport has been reported for the L253P mutation[27], and these transport defects may disrupt arborization and contribute to SCA5 pathogenesis. Disrupted transport may be secondary to defects in microtubule tracts that are organized by the spectrin-actin cytoskeleton[28–31], and/or result from the direct impact of high-affinity binding of L253P β-III-spectrin to the actin-related protein, ARP1[32], a component of the dynactin complex that facilitates cargo transport by microtubule motor proteins[33, 34].

Collectively, the 6.9 Å cryo-EM structure, binding studies, and DEER distance distributions converge on a structural mechanism for disease. The CH2 domain-localized L253P mutation perturbs a closed-open structural equilibrium in β-III-spectrin's ABD by lowering the energetic barrier between structural states. The ABD is then relieved of its regulatory mechanism allowing for the CH1 domain to interact with actin filaments, aided by an additional N-terminal unstructured region that becomes helical upon binding actin.

## Methods

**Protein purification.** For cryo-EM analyses, the WT or L253P human β-III-spectrin ABD coding sequences contained in pET-30a-ABD WT or pET-30a-ABD L253P vectors were expressed in *Escherichia coli* (*E. coli*) BL21(DE3) (Novagen). ABD proteins were both purified using a HiTrap Q 5 mL ion-exchange column followed by a Superdex 200 size exclusion column (GE Healthcare Life Sciences)[7]. Elution fractions of the Superdex 200 column containing pure ABD proteins as assessed by SDS-PAGE were pooled and concentrated (Amicon Ultra-4 Centrifugal Filter, 10 K MWCO). A Bradford assay (Biorad) was then used to determine protein concentrations which equaled 44.0 and 40.6 µM for WT and L253P, respectively. ABD proteins were stored on ice until preparation of ABD–F–actin complexes.

To test the contribution of the N-terminus to actin-binding affinity, the coding sequences for WT ABD (amino acids 1–284) or truncated WT ABD (amino acids 52–284) were PCR amplified using the forward primer AAACACCTGCAAAAAG GTATGAGCAGCACGCTGTCACCC or AAACACCTGCAAAAAGGTGCAGAT GAACGAGAAGCTGTGC and reverse primer AAATCTAGACTACTTCATCTT GGAGAAGTAATGGTAGTAAG. PCR products were digested with AarI and Xba1 restriction enzymes and ligated into the BsaI site of pE-SUMOpro (LifeSensors) containing His and SUMO tags. The final constructs pE-SUMO-ABD WT and pE-SUMO-A52-ABD WT were sequence verified and transformed into *E. coli* BL21 (DE3)pLysS (Agilent). Transformed bacteria were incubated with

rotation at 27 °C in flasks containing 1 L LB media with 100 µg per mL ampicillin and 50 µg per mL chloramphenicol until an absorbance of 0.5 at 550 nm was reached. Then flasks were placed in ice for 10 min before addition of IPTG to 0.5 mM final. The flasks were then incubated with rotation for 4 h in a 22 °C water bath. Bacteria were harvested at 5000 × g and pellets stored at −20 °C. Bacteria were lysed with lysozyme (Sigma) for 1 h at 4 °C in buffer containing 50 mM Tris, pH 7.5, 300 mM NaCl, and 25% sucrose with protease inhibitors (Complete Protease Inhibitor tablet, EDTA-free, Roche), followed by a freeze-thaw cycle using an isopropanol-dry ice bath. Then MgCl2 to 10 mM final and DNase1 (Roche) to 7.5 U per mL final were added and lysate incubated for 1 h at 4 °C. Lysate was clarified at 40,000 × g at 4 °C for 30 min. Supernatants were collected and passed through a 0.45 µm syringe filter before loaded onto a Poly-Prep (Biorad) chromatography column containing 1 mL Ni-NTA agarose (Qiagen) equilibrated in buffer containing 50 mM Tris, pH 7.5, 300 mM NaCl, and 20 mM imidazole. The column was washed with buffer containing 50 mM Tris, pH 7.5, 300 mM NaCl, and 20 mM imidazole, and proteins eluted in buffer containing 50 mM Tris, pH 7.5, 300 mM NaCl, and 150 mM imidazole. Fractions containing ABD proteins were pooled and loaded into a Slide-a-Lyzer, 10 K MWCO, dialysis cassette (ThermoScientific), and dialysis performed at 4 °C in buffer containing 25 mM Tris, pH 7.5, 150 mM NaCl and 5 mM β-mercaptoethanol. To cleave off the SUMO tag, Ulp1 SUMO protease was added to dialyzed ABD proteins at a 1:14 (protease:ABD) mass ratio, and digests incubated for 1.5 h at 4 °C. To separate ABD proteins from the cleaved His-SUMO tag and His-tagged SUMO protease, ABD proteins were loaded onto a Poly-Prep chromatography column containing 0.5 mL Ni-NTA agarose equilibrated in 25 mM Tris, pH 7.5, 150 mM NaCl, and 5 mM β-mercaptoethanol. Elution fractions containing ABD proteins were collected and then loaded onto a gel filtration column (Sephadex S100, GE) equilibrated in buffer containing 10 mM Tris, pH 7.5, 150 mM NaCl, 2 mM MgCl2, and 1 mM DTT at 4 °C. Fractions were analyzed by SDS-PAGE and Coomassie blue staining, and fractions enriched with ABD proteins were pooled and concentrated (Amicon Ultra-15 Centrifugal Filter, 10 K MWCO).

For DEER analyses, the WT and L253P ABD proteins were modified to substitute a serine residue in place of cysteine 115. Spin labeling was performed using native cysteines of which the β-III-spectrin ABD contains four (C76, C115, C186, C231), and C76, C115, and C231 are all solvent exposed. Residues C76 and C231 are best suited for DEER distance measurements, so C115 was mutated to serine to prevent non-specific labeling at that site. The C186 site did not require mutagenesis because it is naturally buried in the core of CH2 and would thus not be accessible to free spin label in solution. PCR site-directed mutagenesis was performed on pET-30a-ABD WT and pET-30a-ABD L253P vector templates using the oligonucleotides CATGCGGATCCACTCCCTGGAGAACGTG and CACGTT CTCCAGGGAGTGGATCCGCATG. The resulting constructs pET-30a-ABD WT C115S and pET-30a-ABD L253P C115S were sequence verified, and then transformed into E. coli BL21(DE3). The ABD C115S proteins were purified as described above. For structural studies of the β-III-spectrin N-terminal residues, a small peptide corresponding to residues SSTLSPTDFDSLEIQGQYSDINNRWDLP DSDWDNDSSSARLFERSRIKALA was produced via solid-state synthesis through Selleck Chemicals LLC.

**Cryo-EM ABD-actin.** 5 µM of rabbit skeletal muscle G-actin was polymerized in 15 mM Hepes-HCl buffer, pH 7.5, 75 mM KCl, 1 mM MgCl2, and 0.5 mM ATP for 2 h at room temperature. For negatively stained samples 2 µM F-actin was incubated with 5–10 µM WT β-III-spectrin or with 2–5 µM β-III-spectrin mutant L253P for 2–20 min. For cryo-samples the mixture (1.5–2 µL) was applied to lacey carbon grids. These grids had previously been plasma cleaned (Gatan Solarus), and were then vitrified in a Vitrobot Mark IV (FEI, Inc.). Images were recorded in a Titan Krios operating at 300 keV, using a Falcon II camera with 1.05 Å per pixel. The images were dose-fractionated into seven "chunks". Each chunk, a sum of multiple frames, represented a dose of ~20 electrons/Å². A total of 586 images (each 4k × 4k) were selected that were free from drift or astigmatism, and had a defocus less than 3.0 µm. The program CTFFIND3[35] was used for estimating the defocus and the range used was from 0.6 to 3.0 µm. The SPIDER software package[36] was employed for almost all further steps in the image processing. The contrast transfer function (CTF) was corrected by multiplying each image by the theoretical CTF, both reversing phases where they need to be reversed and improving the signal-to-noise ratio (SNR). This operation is simply a Wiener Filter in the limit of a very SNR. The program e2helixboxer within the EMAN2[37] suite of programs was used for cutting long filaments from the micrographs. Overlapping boxes, each 384 px long with a 40 px shift between adjacent boxes (~1.5 times the axial rise per subunit) were extracted from these long filaments, yielding ~60,000 segments that were padded to 384 × 384 px. The CTF estimation and particle picking came from the integrated images (containing all seven chunks), while the images used for the initial alignments and reconstruction came from only the first two chunks.

An initial reconstruction using the IHRSR method[38] showed clear decoration of actin, but the mass density due to β-III-spectrin was lower than that from the actin. This appeared to arise from incomplete occupation. We therefore used atomic models of pure actin and actin decorated with α-actinin[39] to sort the segments. Only approximately one-third of the segments showed a higher cross-correlation with the decorated filament, and 20,340 segments were used for further processing. The IHRSR cycles converged to a rotation of −166.9° and an axial rise of 27.3 Å per

subunit. After excluding segments with a large out-of-plane tilt or poor orientation against the reference, 12,443 segments were used in the final reconstruction.

**Model building.** An actin-spectrin asymmetric unit was segmented from the filament map in Chimera[40]. Model building began by docking cryo-EM structure of actin (5JLH)[41] and a predicted model of spectrin CH1 domain into the experimental density data. This predicted spectrin model was generated by I-TASSER[42] based on a crystal structure of plectin (1MB8)[43]. Then the actin–spectrin complex was rebuilt with RosettaCM protocols[44]. A total of 1500 models were generated, and the 10 best models (selected based on Rosetta's energy function) were combined into one model by manual editing in Coot[45] to yield the best overall fit to the density map. A filament model was subsequently generated from this and refined by Phenix real-space refinement[46]. MolProbity[47] was used to evaluate the quality of the model (Supplementary Table 1). The MolProbity scores for the actin-spectrin filament models compare favorably (99th percentile) with structures of similar resolution.

Although segments were sorted to exclude naked actin, Phenix refinement of the actin-spectrin reconstruction clearly shows that the occupancy by spectrin is not 100%, and the actual occupancy is ~75%. Therefore, the threshold chosen for the filament needed to show the full volume for spectrin shows a somewhat larger and lower resolution actin.

**Co-sedimentation assays.** Actin was purified from acetone powder derived from the psoas muscle of New Zealand white rabbit (Oryctolagus cuniculus)[48]. Acetone powder was hydrated in 4 °C water for 30 min to extract actin. The resultant slurry was passed through a Whatman filter paper and 30 mM KCl was added to the filtrate to polymerize actin for a period of 1 h at room temperature. Filamentous actin was then pelleted by 30 min centrifugation at 80,000 rpm in a TLA 100.3 rotor. The actin pellet was resuspended in buffer containing 5 mM Tris, pH 7.6, 0.5 mM ATP, 0.2 mM MgCl2, and homogenized on ice. A 10 min clarifying spin at 70,000 rpm was then performed to pellet aggregate proteins. The G-actin containing supernatant was then isolated and polymerization was initiated by addition of 2 mM MgCl2 with a 30 min incubation at room temperature. The purified ABD proteins were clarified at 100,000 × g at 4 °C for 30 min prior to setting up binding assays. A Bradford assay was performed to determine F-actin and clarified ABD protein concentrations. Binding assays were performed in F-buffer containing 10 mM Tris, pH 7.5, 150 mM NaCl, 0.5 mM ATP, 2 mM MgCl2, and 1 mM DTT. Binding reactions contained 3 µM ABD protein and F-actin ranging from 0 to 140 µM in a total volume of 60 µL. Binding reactions were incubated at room temperature (23–24 °C) for 30 min to reach equilibrium, and then F-actin pelleted by centrifugation at 100,000 × g at 25 °C for 30 min. Unbound ABD was measured by combining 45 µL of binding reaction supernatant with 15 µL 4× Laemmli sample buffer and performing SDS-PAGE followed by Coomassie blue staining. After destaining, gels were scanned using the 700 nm channel in an Odyssey Imager (LI-COR Biosciences). The fluorescence intensities of the ABD protein bands were quantified in Image Studio Lite Ver 5.2 software (LI-COR Biosciences)[49]. Individual ABD band fluorescence intensities were converted to amount ABD protein. This conversion was performed using a standard curve generated by linear regression (Prism 5 software) on ABD Coomassie blue fluorescence intensity values attained from a SDS-PAGE gel loaded with varying amounts of ABD in F-buffer. To determine the dissociation constant ($K_d$) value, data were fit by non-linear regression in Prism 5 software to the equation:

$$Y = X/(K_d + X), \tag{1}$$

where $Y$ equals fraction ABD bound and $X$ equals free F-actin concentration.

**Circular dichroism.** ABD proteins were clarified at 100,000 × g for 20 min at 4 °C. A Bradford assay was performed to determine ABD protein concentrations, and ABD proteins were diluted to 250 ng/µL in buffer containing 10 mM Tris, pH 7.5, 150 mM NaCl, 2 mM MgCl2, 1 mM DTT. CD spectra were acquired in a Jasco J-815 Spectropolarimeter equipped with a Peltier temperature controller. Immediately before analysis, the instrument was baseline-corrected using ABD protein buffer. For secondary structure analyses, CD spectra were measured from 200 and 260 nm at 25 °C. Thermal unfolding of the ABD protein sample was analyzed by recording CD spectra at 222 nm over the temperature range of 20–85 °C. CD analyses were performed three times for each ABD protein. Non-linear regression analysis was performed in Prism 5 (GraphPad Software, Inc.) to determine the melting temperature using the following equation for a two-state transition, reported previously[50].

$$Y = (\alpha_N + \beta_N T)/\{1 + e^{4T_m(T-T_m)/T\Delta T}\} + (\alpha D + \beta_D T)/\{1 + e^{4T_m(T_m-T)/T\Delta T}\}, \tag{2}$$

where $Y$ is the CD signal at temperature $T$, $T_m$ is melting temperature, $\Delta T$ is the width of the unfolding transition, $\alpha_N$ and $\alpha_D$ are the intercepts of the native and denatured states, respectively, and $\beta_N$ and $\beta_D$ are the slopes of the native and denatured states, respectively.

For secondary structure analysis of the N-terminal peptide, a lyophilized powder was reconstituted in the same buffer system as that described for the ABD

proteins. The reconstituted peptide was then diluted to a concentration of 99 ng per µL and subsequently scanned over the same wavelength range described above. In the case of both ABDs and peptide, raw ellipticity was normalized to each sample's respective concentrations according to the following equation:

$$MRE = [\theta(MW/N - 1)]/(lc), \qquad (3)$$

where $\theta$ represents the raw ellipticity, MW represents the protein molecular weight, $N$ is the number of amino acids, $l$ is the path length, and $c$ is the concentration in milligrams per milliliter.

**Statistical analyses**. Unpaired, two-tailed $t$-tests were performed in Prism 5 software to determine whether significant differences existed in ABD protein melting temperatures or 222/208 absorbance ratios determined by CD. The $n$ value was equal to three in all cases.

**Spin labeling**. In the β-III-spectrin ABD constructs, 500 µM of the spin label 4-maleimido-TEMPO (MSL, 4-maleimido-2,2,6,6-tetramethyl-1-piperidinyloxy; Sigma-Aldrich) was added to 25 µM protein and equilibrated on a rocker for 3 h at 4 °C. Prior to addition of MSL, the protein solution had been run over a Zeba desalting column pre-equilibrated with 10 mM Tris, pH 7.5, and 150 mM NaCl to remove most of the 1 mM DTT left over from size exclusion. After the spin-label incubation period, the protein was once again subjected to a Zeba desalting column to remove any unreacted spin label. To ensure complete removal, however, the spin-labeled protein was then subjected to three 4 h rounds of dialysis in 4 L solutions contain 10 mM Tris, pH 7.5, 150 mM NaCl, and 1 mM DTT. MSL was ultimately chosen over the more commonly used (1-oxyl-2,2,5,5,-tetra-methylpyrroline-3-methyl) methanethiosulfonate (MTSSL) because spin labeling of the β-III-spectrin ABD constructs was incomplete, requiring inclusion of DTT reducing agent post-labeling to prevent undesired ABD cross-linking. Incubation of the ABD constructs with spin label for periods longer than 3 h resulted in significant protein loss due to precipitation. The spin-labeled WT and L253P β-III-spectrin ABD constructs were concentrated down to 230 and 175 µM, respectively, prior to spin counting and DEER sample preparation.

**EPR spectroscopy**. To verify labeling, a continuous wave EPR spectrum was acquired with sample temperature of 296 K on the E500 Bruker EPR spectrometer operating at X-band (9.5 GHz) and equipped with an SHQ cavity. The derivative spectrum was then doubly integrated to determine spin concentration by comparing with the double integral of a 100 µM TEMPOL standard (Supplementary Fig. 5b, c). For WT and L253P β-III-spectrin ABD constructs, spin concentrations were determined to be 98 and 75 µM, respectively, indicating a labeling efficiency of ~43% for both. After spin counting, we performed DEER on β-III-spectrin ABD constructs doubly labeled with MSL. ABD samples were prepared by adding 7% v/v glycerol (as a cryoprotectant), loading samples into quartz capillaries (1.1 mm i.d., 1.6 mm o.d., 15 µL sample volume) and subsequently flash freezing samples in liquid nitrogen after which samples were stored at −80 °C until use. A Bruker E580 spectrometer operating at Q-band (34 GHz) with an EN5107 resonator was then used to implement a four-pulse DEER protocol with a $\pi/2$ pulse width of 12 ns and an electron double resonance (ELDOR) pulse width of 24 ns[51]. The ELDOR frequency was placed on the pump position which corresponded to the absolute maximum of the nitroxide absorption spectrum. The observe position was placed 24 Gauss higher than the pump position on the field swept absorption spectrum. Experiments were run at a temperature 65 K. After data acquisition, background-corrected DEER decays were analyzed using the Tikhonov regularization method provided in DeerAnalysis2013.2 to extract distance distributions encoded in the waveform (Supplemental Fig. 6)[52]. To determine the stable components in the resulting Tikhonov distributions, we examined the impact of a range background models whose starting evolution time varied from 0.5 to 2.4 µs. The components that were invariable were used for structural interpretation of the ABD constructs. The component that was not stable (the peak at >7.0 nm) was excluded from structural interpretation as it represents an artifact of background subtraction.

**Data availability**. The reconstruction was deposited in the Electron Microscopy Data Bank with accession number 8886 and the corresponding filament model was deposited in the Protein Data Bank with accession number 6ANU. Other data are available from the corresponding author upon reasonable request.

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

## Acknowledgements

This work was partially supported by grants from NIH to T.S.H. (GM44757), D.D.T. (AG32961), and E.H.E. (GM81303), and from NSF (MCB-1616854 to D.D.T.). A.W.A. was supported by an NIH grant (GM44757) to T.S.H. and an NIH training grant to D.D.T. (T32 AR007612). M.E.F. was supported by an NIH training grant to D.D.T. (T32 AR007612). The cryo-EM work was conducted at the Molecular Electron Microscopy Core facility at the University of Virginia, which is supported by the School of Medicine and built with NIH grant G20-RR31199. The Titan Krios and Falcon II direct electron detector within the Core were purchased with NIH SIG S10-RR025067 and S10-OD018149, respectively.

## Author contributions

A.W.A. and T.S.H. initiated the project. A.W.A. and M.E.F. in consultation with D.D.T. and T.S.H. designed the experiments, excluding the cryo-EM and structural modeling that were conducted by F.W., A.O., and E.H.E. A.W.A. and M.E.F. expressed, purified, and characterized all protein samples. A.W.A., M.E.F., A.R.T, and D.D.T. analyzed DEER data. The manuscript was prepared by A.W.A. and M.E.F. with edits by D.D.T., E.H.E., and T.S.H.

## Additional information

**Competing interests:** The authors declare no competing financial interests.

