## [Peer Review file · Nature Communications]

Reviewers' comments:

Reviewer #1 (Remarks to the Author):

The authors tried to elucidate the mechanism of a neurodegenerative disease called spinocerebellar ataxia type 5, which is caused by a mutation in the cytoskeletal protein β -III-spectrin. They used cryo-electron microscopy to visualize the structure of actin filament complexed with the N-terminal actin-binding domain (ABD) of β -III-spectrin with the L253P mutation, which causes a 1000-fold increase in actin binding affinity. They used two calponin homology domains (CH1 and CH2) of the ABD for this structural analysis and identified that the CH1 domain and the N-terminal α -helix extending from it are both bound tightly to the surface of actin filament in the density map at 7 Å resolution. They confirmed the role of this N-terminal α -helix for the binding stability of the ABD by truncating the N-terminal 51 residues. They also used EPR spectroscopy to measure the inter-CH domain distance of the ABD in solution and showed that the L253P mutation increases the fraction of its open state in which the surface of CH1 used to bind to actin filament is exposed to solution.

This is an interesting study, and the experimental results appear to be solid. The biochemical and physicochemical data are of high quality, supporting the conclusion of the paper. However, the resolution of the cryo-EM structural analysis of the ABD-bound actin filament is rather limited as a recent standard, and it is difficult to judge the quality of the 3D density map and model fitting due to a poor representation of the map and model in Figure 1. The density for the N-terminal α -helix of the ABD is especially difficult to recognize in Figure 1b because of the faint appearance of the density map. The silhouette of the map should be more emphasized for better visualization. The authors stated that they had to mask the ABD segment to amplify its density level by 30% because the reconstructed density map showed a relatively low-density level for the ABD compared to that of actin. But how they masked the ABD segment is not described in Materials and Methods. This kind of procedure is dangerous because it could generate an artificial amplification of a specific density segment with a model bias depending on the way it is done.

It is clear from the data that the high-affinity binding of β -III-spectrin ABD to actin filament is caused by the L253P mutation that increases the fraction of the ABD in its open state. However, no discussion is given as to how this high affinity binding results in the neurodegenerative disease. Implications of the results should be discussed in a more comprehensible manner for general readership.

There is a statement that the N-terminal sequence of the ABD is critical to binding ... potentially indirectly through allosteric destabilization of the ABD. Is there any experimental evidence for this?

Minor points:

1. Supplementary Figure 2: A gold-standard FSC should also be presented. Even if the number of image segments is small, it should be possible to produce it.
2. Supplementary Figure 3: The color of F-actin is grey, not cyan as indicated in the legend. It would be nice to have a different view of the model to see the clash between calmodulin and F-actin.
3. Supplementary Figure 5: CH1 and CH2 should be labeled.

Reviewer #2 (Remarks to the Author):

In this brief communication the authors use both cryo-EM and DEER spectroscopy to investigate the structural consequences of the L253P mutation in the actin-binding domain of β -III-spectrin. The L253P mutation is known to cause a ~1000-fold increase in binding affinity between β -III-

spectrin and actin resulting in spinocerebellar ataxia type 5. As requested, this review will focus on the DEER data and its analysis.

The authors use DEER spectroscopy to measure the distance distribution between a spin label in CH1 and one in CH2 both in the WT and the L253P forms of the ABD of β -III-spectrin. The authors use naturally occurring cysteines in CH1 (76) and CH2 (231) for labeling with 4-maleimido-tempo.

No explanation is given for using a maleimide rather than the more commonly used methanethiosulfonate label but presumably it is due to the need to maintain β -III-spectrin in DTT.

The authors present CW-EPR and spin counting data for the two spin labeled proteins but do not give corresponding measured estimates of the protein concentrations that would be needed to assess the labeling efficiency. Based on the DEER data, it appears that the labeling efficiency is rather low resulting in a small modulation depth and thus a lower S/N in the background corrected data (Supp. Fig. 6, bottom row). What is the labeling efficiency obtained for these samples?

While it is likely that the data could be improved with increased labeling efficiency, there are subtle changes in the data sets that are consistent with a significant structural change as a result of the L253P mutation. The two DEER data sets, WT and L253P, were analyzed using Tikhonov regularization in DeerAnalysis 2013.2. The L-curves suggest that the optimal regularization parameter was well-defined in both cases. The authors then fit the distance distributions obtained from the Tikhonov analysis to a sum of Gaussians. While it is true that fitting of a distance distribution obtained from fitting the primary DEER data has been used by others (no appropriate references are given), this approach is questionable under the best of circumstances and inappropriate under these. If a description of the interspin distance distribution in terms of a sum of Gaussians is desired, methods for fitting DEER data directly to such a model do exist. However, in my opinion, for this communication there is no compelling reason to reduce the distance distributions obtained from Tikhonov analysis to a sum of Gaussians.

Analysis of the WT DEER data gives a large component centered near the distance expected based on the homology model in Supp. Fig. 5 and a smaller component at a considerably longer distance ($> 70 \text{ \AA}$). The results in Supp. Fig. 6 make clear that this 2nd component is needed to adequately describe the distance distribution obtained from fitting the DEER data but it is not included in the distributions presented in Fig. 3 of the communication. Is this 2nd component included in the fit to the WT DEER data presented in Fig. 3A or not? No justification is given for ignoring this 2nd component and it is not consistent with the authors conclusion that the L253P favors an open state more than WT.

The plot legend for the L253P data (Supp. Fig 6, 3rd row) states that 3 Gaussians are needed to adequately describe the data even though the RMSD plot (Supp. Fig 6, 3rd row) suggests that either 2 or 4 Gaussians might be a better choice. How is the number of Gaussians needed to adequately describe the data determined? What do the horizontal dashed lines in the 3rd row of Supp. Fig. 6 indicate? The fits to the data in the bottom row of Supp. Fig. 6 are actually not very good, particularly for the L253P. Are the original fits obtained using Tikhonov regularization significantly better than those presented?

Often Q-band DEER data contains an artifact at longer time values. Was the data truncated to remove such artifacts?

Given these serious concerns about the analysis of the DEER data and the conclusions drawn I cannot recommend this manuscript for publication in Nature Communications. In particular, there is a significant long-distance component in the WT DEER data which is ignored in the main text and contradicts the authors conclusion that the ABD of β -III-spectrin undergoes an opening as a result of the L253P.

Reviewer #3 (Remarks to the Author):

The paper describes important novel findings that are of wider interest. This includes the following original conclusions:

- Evidence that the β -III-spectrin's ABD binds to F-actin via the CH1 domain. The N-terminal unstructured region of CH1 is shown to become helical upon binding actin, thereby strengthening

the interaction between the proteins.

- Evidence that CH2 is not contributing to F-actin binding.
- The CH2 domain-localized L253P spinocerebellar ataxia type 5 mutation is shown to perturb the closed-open equilibrium between the β -III-spectrin's CH1 and CH2 domains by lowering the energetic barrier between structural states.

The results are well-presented and the data support all conclusions. I feel that this paper will have a considerable impact in the field.

Data analysis appears to be appropriate and valid. The information provided with the supplemental material is sufficiently detailed for reproduction of the work.

Minor comments:

In its current form, the manuscript is well-accessible to experts in the field but it contains a couple of potential stumbling blocks for students or biomedical researchers working in different fields. Here are a couple of suggested changes.

Start by describing the domain structure of the ABD as composed of two CH domains. Be clear about the assignment of the N-terminal unstructured region as part of CH1.

Give reason for the C115S substitution used for the DEER analysis in the main text.

The use of Coomassie blue as near-infrared fluorescent stain is not common practice in most biochemistry laboratories and requires a reference to the original work describing the technique (Butt R.H., Coorssen J.R. Coomassie blue as a near-infrared fluorescent stain: A systematic comparison with Sypro Ruby for in-gel protein detection. *Mol. Cell. Proteom.* 2013;12:3834–3850. doi: 10.1074/mcp.M112.021881)

Dietmar J. Manstein

The authors would like to thank the reviewers for their critical analysis of the manuscript and suggested changes. We believe the revised version is significantly improved as a result and hope it is found suitable for continued consideration. Below is a point-by-point description of how each reviewer critique has been addressed. The original review comments are presented with **black** text whereas our responses are presented with **red** text. Additionally, corresponding changes to the manuscript or supplemental materials documents are highlighted and color-coded according to reviewer: reviewer 1 changes are highlighted **yellow**; reviewer 2 changes are highlighted **cyan**; reviewer 3 changes are highlighted **green**.

REVIEWER 1

Concern #1 – Poor representation of the cryo-EM density map

This is an interesting study, and the experimental results appear to be solid. The biochemical and physicochemical data are of high quality, supporting the conclusion of the paper. However, the resolution of the cryo-EM structural analysis of the ABD-bound actin filament is rather limited as a recent standard, and it is difficult to judge the quality of the 3D density map and model fitting due to a poor representation of the map and model in Figure 1. The density for the N-terminal α -helix of the ABD is especially difficult to recognize in Figure 1b because of the faint appearance of the density map. The silhouette of the map should be more emphasized for better visualization.

We have emphasized the silhouette and the N-terminal helix is now more prominent. We show part of the new Fig. 1 to the right. The complete figure is now shown in the revised manuscript.

Concern #2 – Possible model bias introduced during amplification of density level

The authors stated that they had to mask the ABD segment to amplify its density level by 30% because the reconstructed density map showed a relatively low-density level for the ABD compared to that of actin. But how they masked the ABD segment is not described in Materials and Methods. This kind of procedure is dangerous because it could generate an artificial amplification of a specific density segment with a model bias depending on the way it is done.

We have now used figures in the paper (such as that shown above) with no amplification of the spectrin density, avoiding any potential model bias. The only downside of this is that the actin density appears to be at lower resolution as the threshold for actin is lower than it should be.

Concern #3 – Need for discussion of how increased affinity potentially contributes to neurodegenerative disease

It is clear from the data that the high-affinity binding of β -III-spectrin ABD to actin filament is caused by the L253P mutation that increases the fraction of the ABD in its open state. However, no discussion is given as to how this high affinity binding results in the neurodegenerative disease. Implications of the results should be discussed in a more comprehensible manner for general readership.

We have broadened the discussion to add speculation about how a greatly increased affinity for actin might lead to neurodegenerative disease. This added material is highlighted yellow in the revised version of the main text.

Concern #4 – Need for literature support of statement about N-terminus allosterically affecting binding affinity in ABDs

There is a statement that the N-terminal sequence of the ABD is critical to binding... potentially indirectly through allosteric destabilization of the ABD. Is there any experimental evidence for this?

After our manuscript was submitted, a paper appeared in *Biochemistry* (Singh et al., 2017) showing how the removal of the N-terminal extension of a spectrin homolog, utrophin, actually *stabilizes* the CH1 domain. This must be an allosteric effect. The new paper explicitly shows how this N-terminal extension greatly increases the affinity of utrophin for actin, and leads to the suggestion that the weaker affinity of dystrophin for actin is due to the lack of this extension in dystrophin. This new manuscript, however, suggests that the N-terminal extension in utrophin never binds actin, as a peptide containing this sequence fails to bind. Thus, this new paper, now discussed in our manuscript (added text is highlighted yellow) and added to the list of references, significantly increases the potential impact of our results.

Minor Concern #1 – In Supplementary Figure 2, a gold-standard FSC should also be presented. Even if the number of image segments is small, it should be possible to produce it.

While we never doubted that it would be possible to produce such an FSC, we urge the reviewer to look at Subramaniam et al., *Current Opinion in Structural Biology* (2016). It is stated there:

“...they highlight the fact that the FSC approach of comparing half-maps is an intrinsically biased measurement, dependent upon the number (N) of particles. This is illustrated in Figure 3e, where we show a theoretical curve for resolution as a function of sample size N, based upon the observations that this curve tends to follow a log-linear relationship [54] and becomes asymptotic at some large value of N, where an increase in N will lead to no further improvement

in resolution. If, when $N = 100\,000$, the resolution is asymptotic and is actually 3.3 \AA for the full map, then the comparison of the two half-maps will yield a resolution of 3.6 \AA (the resolution for 50,000 particles or helical segments). For a map that has 40,000 particles or segments, a resolution of 3.8 \AA for the full map will show a resolution of 5.3 \AA when comparing two half-maps, significantly worse than the resolution of the full map that we are trying to estimate.”

Nevertheless, we have generated a “gold-standard” FSC (shown here) which yields a resolution of 6.9 \AA . We have included both, completely independent estimates of the resolution in Supplemental Figure 2.

Minor Concern #2 – In Supplementary Figure 3, the color of F-actin is grey, not cyan as indicated in the legend. It would be nice to have a different view of the model to see the clash between calmodulin and F-actin.

In the revised Supplementary Figure 3 (below), we have fixed the legend and added a different view to better see the clash. The added view is on the right in the figure below. It is now much clearer how calmodulin (orange) would be clashing with actin (gray). This revised figure now appears in the Supplemental Materials document.

Minor Concern #3 – In Supplementary Figure 5, CH1 and CH2 should be labeled.

We have done this.

REVIEWER 2

Concern #1 – No rationale provided for choice of spin label

No explanation is given for using a maleimide rather than the more commonly used methanethiosulfonate label but presumably it is due to the need to maintain β -III-spectrin in DTT.

The reviewer correctly deduced the need for use of MSL. Our buffering conditions included DTT reducing agent. With lower labeling efficiencies observed for these protein constructs (see below), DTT post-labeling prevented any potential ABD cross-linking.

Additional text summarizing this has been added to the Supporting Materials and Methods in the subsection titled “Spin labeling” and is highlighted cyan.

Concern #2 – Details about labeling efficiency missing

The authors present CW-EPR and spin counting data for the two spin labeled proteins but do not give corresponding measured estimates of the protein concentrations that would be needed to assess the labeling efficiency. Based on the DEER data, it appears that the labeling efficiency is rather low resulting in a small modulation depth and thus a lower S/N in the background corrected data (Supp. Fig. 6, bottom row). What is the labeling efficiency obtained for these samples?

The protein concentrations for WT and L253P ABD constructs were 230 μM and 175 μM , respectively. Their corresponding spin counts were 98 μM and 75 μM , generating similar labeling efficiencies of $\sim 43\%$. This low efficiency was observed in spite of a 10-fold molar excess of probe to cysteine residues. Longer equilibration periods for labeling resulted in a loss of protein due to precipitation. Additional text summarizing this has been added to the Supporting Materials and Methods in the subsections titled “Spin labeling” and “EPR spectroscopy” and is highlighted in cyan.

Concern #3 – Inappropriate fitting of Tikhonov distance distributions to multiple Gaussian functions

The authors then fit the distance distributions obtained from the Tikhonov analysis to a sum of Gaussians. While it is true that fitting of a distance distribution obtained from fitting the primary DEER data has been used by others (no appropriate references are given), this approach is questionable under the best of circumstances and inappropriate under these. If a description of the interspin distance distribution in terms of a sum of Gaussians is desired, methods for fitting DEER data directly to such a model do exist. However, in my opinion, for this communication there is no compelling reason to reduce the distance distributions obtained from Tikhonov analysis to a sum of Gaussians.

Following this suggestion, we have removed the Gaussian distributions and present only the Tikhonov distance distributions. This makes presentation of our results more straightforward to the reader. This change addresses a majority of the original concerns summarized in “Concern #4” below. The one concern it does not address regarding distributions at a longer distance >7.0 nm is addressed below under “Concern #6.”

Concern #4 – Unclear presentation of Gaussian fitting of Tikhonov distance distribution

Analysis of the WT DEER data gives a large component centered near the distance expected based on the homology model in Supp. Fig. 5 and a smaller component at a considerably longer distance (> 70 Å). The results in Supp. Fig. 6 make clear that this 2nd component is needed to adequately describe the distance distribution obtained from fitting the DEER data but it is not included in the distributions presented in Fig. 3 of the communication. Is this 2nd component included in the fit to the WT DEER data presented in Fig. 3A or not? No justification is given for ignoring this 2nd component and it is not consistent with the authors conclusion that the L253P favors an open state more than WT. The plot legend for the L253P data (Supp. Fig 6, 3rd row)

states that 3 Gaussians are needed to adequately describe the data even though the RMSD plot (Supp. Fig 6, 3rd row) suggests that either 2 or 4 Gaussians might be a better choice. How is the number of Gaussians needed to adequately describe the data determined? What do the horizontal dashed lines in the 3rd row of Supp. Fig. 6 indicate? The fits to the data in the bottom row of Supp. Fig. 6 are actually not very good, particularly for the L253P. Are the original fits obtained using Tikhonov regularization significantly better than those presented?

These concerns have been addressed by reporting only the Tikhonov distributions. All of the text dedicated to Gaussian fitting of the Tikhonov distributions in the “EPR spectroscopy” subsection of the Supporting Materials and Methods has been removed. The Gaussian fitting analysis originally presented in Supplemental Figure 6 has been removed and replaced by just Tikhonov distance distributions.

Concern #5 – Question of whether or not DEER data was terminally truncated to remove data artifact that occurs at longer evolution time values

Often Q-band DEER data contains an artifact at longer time values. Was the data truncated to remove such artifacts?

Yes, the data sets were truncated. This artifact is often present in X-band DEER as well and is due to some excitation bandwidth overlap when the pump and observe pulses get close to each other at the end of the dataset.

Concern #6 – Apparent long distance at $>70\text{\AA}$ contradicts proposed structural model for the beta spectrin ABD

In particular, there is a significant long-distance component in the WT DEER data which is ignored in the main text and contradicts the authors conclusion that the ABD of β -III-spectrin undergoes an opening as a result of the L253P.

The $>70\text{\AA}$ distance in the Tikhonov is an artifact of background subtraction which becomes apparent when examining overlaid distance distributions derived from different background choices. As now illustrated in Supplementary Figure 6 (part of which is shown below), the 4.8 nm distance (which correlates with the predicted distance in the homology model), is robust and minimally fluctuates irrespective of background choice. This contrasts with the >7.0 nm distance which appears as an unstable peak characterized by dramatic fluctuations in amplitude.

Ultimately, the background chosen for the Tikhonov used in the main text resulted from the background model with the best RMSD. In Supp. Fig. 6, the panels originally allocated to showing Gaussian fits to the Tikhonov distance distributions have now been replaced with overlaid Tikhonov distributions derived from distinct background models to illustrate stable and unstable populations. Additionally, these panels contain insets showing the RMSD as a function of background start which was varied from 0.5-2.4 μ sec. The Supp. Fig. 6 figure legend also now includes a summary description for identifying stable and unstable populations and how that, in turn, identifies the >7.0 nm distance as an artifact that results in its exclusion from structural interpretation of the β -III-spectrin ABDs. Because the artifact peak is excluded from structural interpretation of the ABD constructs, it is not shown as part of the Tikhonov distance distribution displayed in the main text. The updated version of Fig. 3 in the main text is shown below and is included in the revised manuscript.

A summary of the Tikhonov analysis provided here has also been added to the Supporting Materials and Methods in the subsection “EPR spectroscopy” and is highlighted in cyan. Text and figure call-outs were also modified to reflect changes to DEER data and are highlighted in cyan.

REVIEWER 3

Concern #1 – Additional information should be added about ABD and N-terminal region structures to improve accessible of introductory material to non-experts

In its current form, the manuscript is well-accessible to experts in the field but it contains a couple of potential stumbling blocks for students or biomedical researchers working in different fields. Here are a couple of suggested changes. Start by describing the domain structure of the ABD as composed of two CH domains. Be clear about the assignment of the N-terminal unstructured region as part of CH1.

We have now added additional text describing the general structure of actin-binding domains and how they contain N-terminal residues of varying length. This added information is highlighted in green in the revised manuscript.

Concern #2 – No rationale is provided for the generation of the C115S mutant ABD that was used for the DEER analysis in the main text

The rationale behind introducing the C115S point mutation was to ensure site-specific labeling. The beta spectrin ABD has 4 native cysteines (C76, C115, C186, C231). C76, C115 and C231 are all solvent exposed, but C76 and C231 seemed best suited for DEER distance measurements. As such, C115 was mutated to serine to prevent labeling at that site. C186 did not require mutagenesis because it is buried in the core of CH2. A similar description for the C115S mutation has now been added to the revised text and is highlighted in green.

Concern #3 – Missing reference for use of Coomassie blue as a near-infrared fluorescent stain

We have now added the reference indicated by this author to the Materials and Methods, specifically in the subsection entitled “Co-sedimentation assays.”

REVIEWERS' COMMENTS:

Reviewer #1 (Remarks to the Author):

The manuscript is improved by the revisions including modifications in figure presentations and additional discussion on the high-affinity actin binding of β -III-spectrin by the L253P mutation. Am I correct that the phenotype of the L253P mutation is not yet known? If anything is known on the biological implication of this mutation, it should be described.

As I mentioned in my previous review report, the resolution of the cryo-EM structural analysis of the ABD-bound actin filament (6.9 Å) is rather limited as a recent standard or expectation by using Titan Krios as a microscope and Falcon II, a direct electron detector, as a camera. In this experimental setting, the resolution is expected to reach below 5 Å or at least below 6 Å. The authors described in Supplementary Material that the occupancy of bound ABD is around 75%, and this low occupancy may be the reason for the limited resolution. The possible reason for the limited resolution should be described in the text.

The presentation in Figure 1 is improved for easier recognition of the map quality and model fit. Now it could be made easier for general readers to recognize the spectrin CH1 domain and four actin subdomains if they are labeled in Figure 1, especially in Figure 1b.

The CH1 domain appears to bind to two actin subunits in a similar manner to myosin head, and the binding surfaces on actin molecules seem to be also shared by them. The similarity and differences between β -III-spectrin CH1 domain and myosin head in F-actin binding should be described briefly by comparing with recent actomyosin structures, such as those by von der Ecken et al (2016) Nature and Fujii and Namba (2017) Nature Communications.

Reviewer #2 (Remarks to the Author):

I am satisfied by the authors responses to my questions regarding the original submission and all of the changes to the manuscript and the supplementary material are appropriate. I recommend publication as is.

Response to Reviewers

Reviewer #1:

1. *The manuscript is improved by the revisions including modifications in figure presentations and additional discussion on the high-affinity actin binding of β -III-spectrin by the L253P mutation. Am I correct that the phenotype of the L253P mutation is not yet known? If anything is known on the biological implication of this mutation, it should be described.*

The questions regarding the phenotype of the L253P β -III-spectrin mutation and its biological implications are helpful. To address these questions we have provided additional text at the beginning of the discussion section that provides better context for the mutation and the implications of our findings.

2. *As I mentioned in my previous review report, the resolution of the cryo-EM structural analysis of the ABD-bound actin filament (6.9 Å) is rather limited as a recent standard or expectation by using Titan Krios as a microscope and Falcon II, a direct electron detector, as a camera. In this experimental setting, the resolution is expected to reach below 5 Å or at least below 6 Å. The authors described in Supplementary Material that the occupancy of bound ABD is around 75%, and this low occupancy may be the reason for the limited resolution. The possible reason for the limited resolution should be described in the text.*

Attaining high resolution (now becoming almost routine) by cryo-EM requires an excellent microscope, a direct electron detector, **and a sample that is highly ordered**. Simply having only two of the three (e.g., a Titan Krios with a Falcon II camera) does not guarantee a resolution better than 6 Å. To see this, we have looked at all structures deposited in the EMDB databank that were imaged on a Titan Krios with either a Falcon II or a Gatan K2 detector (since the K2 has been used for the majority of the highest resolution cryo-EM structures). We find 245 depositions with a resolution of 7 Å or worse! Of these 98 are at a resolution of 10 Å or worse. We have excluded in this search all cryo-EM tomography which will be at a lower resolution, and are looking at only single-particle and helical reconstructions. These depositions include many recent papers, such as Zhang et al., Cell (2017) where the resolutions were 8.7 and 15 Å:

Zhang, K., Foster, H.E., Rondelet, A., Lacey, S.E., Bahi-Buisson, N., Bird, A.W., and Carter, A.P. (2017). Cryo-EM Reveals How Human Cytoplasmic Dynein Is Auto-inhibited and Activated. Cell 169, 1303-1314 e1318.

Risi et al., PNAS (2017) where the resolutions were 8 and 11 Å:

Risi, C., Eisner, J., Belknap, B., Heeley, D.H., White, H.D., Schroder, G.F., and Galkin, V.E. (2017). Ca²⁺-induced movement of tropomyosin on native cardiac thin filaments revealed by cryoelectron microscopy. Proc Natl Acad Sci U S A 114, 6782-6787.

and Ozorowski et al., Nature (2017) where the resolutions were 8.6 and 10.5 Å:

Ozorowski, G., Pallesen, J., de Val, N., Lyumkis, D., Cottrell, C.A., Torres, J.L., Copps, J., Stanfield, R.L., Cupo, A., Pugach, P., et al. (2017). Open and closed structures reveal allostery and pliability in the HIV-1 envelope spike. Nature 547, 360-363.

We have no reason to believe that the 75% occupancy is the source of the limited resolution. The limitation on resolution is simply the structural coherence within the segments that we are using for the single-particle approach to helical reconstruction. This is why some helical polymers may be solved

at a resolution of 3.2 Å, while using the same microscope and camera other polymers may only reach a resolution of 7 Å, 10 Å, or even worse.

3. The CH1 domain appears to bind to two actin subunits in a similar manner to myosin head, and the binding surfaces on actin molecules seem to be also shared by them. The similarity and differences between β -III-spectrin CH1 domain and myosin head in F-actin binding should be described briefly by comparing with recent actomyosin structures, such as those by von der Ecken et al (2016) Nature and Fujii and Namba (2017) Nature Communications.

We were puzzled by this comment, as we see no significant similarities between the binding of spectrin's CH1 and the binding of myosin to F-actin. We show below the structure of actomyosin from von der Ecken, with our structure of spectrin superimposed. Given the lack of similarity, we think that a discussion of this comparison would be confusing to the reader. If we were to discuss the differences, then why not discuss the differences with other F-actin binding proteins such as vinculin, cofilin, ADF, SipA, etc.?

4. The presentation in Figure 1 is improved for easier recognition of the map quality and model fit. Now it could be made easier for general readers to recognize the spectrin CH1 domain and four actin subdomains if they are labeled in Figure 1, especially in Figure 1b.

The extra labeling has been added to Figure 1b as requested.